# Guiding Multi-Step Rearrangement Tasks with Natural Language Instructions

**Elias Stengel-Eskin**[1][*]    **Andrew Hundt**[1][*]   **Zhuohong He**[1]    **Adit Murali**[1]

**Nakul Gopalan**[2]    **Matthew Gombolay**[2]    **Gregory Hager**[1]
[1]Johns Hopkins University        [2] Georgia Institute of Technology

**Abstract:** Enabling human operators to interact with robotic agents using natural language would allow non-experts to intuitively instruct these agents. Towards this goal, we propose a novel Transformer-based model which enables a user to guide a robot arm through a 3D multi-step manipulation task with natural language commands. Our system maps images and commands to masks over grasp or place locations, grounding the language directly in perceptual space. In a suite of block rearrangement tasks, we show that these masks can be combined with an existing manipulation framework without re-training, greatly improving learning efficiency. Our masking model is several orders of magnitude more sample efficient than typical Transformer models, operating with hundreds, not millions, of examples. Our modular design allows us to leverage supervised and reinforcement learning, providing an easy interface for experimentation with different architectures[2]. Our model completes block manipulation tasks with synthetic commands 530% more often than a UNet-based baseline, and learns to localize actions correctly while creating a mapping of symbols to perceptual input that supports compositional reasoning. We provide a valuable resource for 3D manipulation instruction following research by porting an existing 3D block dataset with crowdsourced language to a simulated environment. Our method's 25.3% absolute improvement in identifying the correct block on the ported dataset demonstrates its ability to handle syntactic and lexical variation.

**Keywords:** Instruction following, object grasping and manipulation, multimodal fusion, computer vision for robotic applications

## 1   Introduction

Reinforcement Learning (RL) is a powerful tool for developing perception-driven robotic systems. Most RL algorithms train an agent to perform a single task based on an given reward function. However, what if we would like to adapt or extend the learned policy of a robot to novel or more specialized tasks by providing instructions? Could we exploit previously learned RL policies for a robot to solve novel tasks *now* specified using natural language? The ability to adapt learned policies through natural language to perform novel tasks would open the door to a modular integration of perception, language, and action, avoiding the complexities of joint training over all modalities simultaneously. For example, in our experiments, we constrain a general-purpose block manipulation policy using natural language, without needing any language data to train the policy itself.

This kind of modular learning requires injecting guidance from language into the RL-driven action policy while simultaneously addressing the challenges of natural language understanding. Natural language semantics is often underspecified, especially with respect to a physical environment; correctly choosing one out of many potentially valid actions for a command such as "Move the green block to the left of the right block" involves pragmatic and common-sense reasoning. Furthermore, in order to map from language to action, we need to address the symbol grounding problem [1], i.e.

---

[*]Equal contribution; {`elias,ahundt`}`@jhu.edu`
[2]Code and data: `https://github.com/esteng/guiding-multi-step`

5th Conference on Robot Learning (CoRL 2021), London, UK.

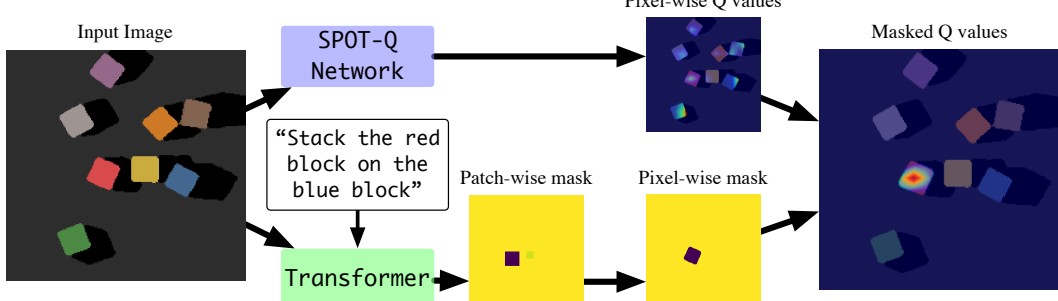

Figure 1: Schematic of our model. From an input image, pixel-wise Q-values are generated using a pre-trained network. Simultaneously, a Transformer-based model produces language-conditioned masks, which are heuristically turned into pixel-wise masks over the image. These masks are combined with the Q-values to find the highest Q-value that respects the language constraints.

how does one map symbols (e.g. "left", "right", "block", "red") to the physical world? We leverage Transformer encoders to *learn* a strategy from data, in place of the manually defined data fusion strategies found in prior work [2, 3, 4, 5, 6, 7, 8].

In our work, we factorize language-driven action into learning *how to act* and learning *where to act*, following Misra et al. [4]. Learning how to act is accomplished by the SPOT-Q algorithm [9] which teaches an agent to re-arrange colored blocks into stacks and rows by grasping and placing them. We use SPOT-Q models pre-trained without any language data for our manipulation experiments, demonstrating that our method lets us leverage existing manipulation models without additional training. SPOT-Q identifies regions worth exploring, *i.e.* containing an object, to encode a set of valid action regions for grasping and placing. We produce segmentation masks that are jointly conditioned on both the language input and an image of the state and apply them with to set of pixel-wise Q-values from the SPOT-Q RL framework. This process ideally provides us with the best action for a given state and command, as Figure 1 illustrates.

Factorizing the learning process this way provides several advantages. Firstly, it offers a more efficient path to learning. Separating the language grounding model from the reinforcement learning model leads to a less complex RL model that receives more direct feedback for tasks. Moreover, the separation of action and language understanding allows us to learn the language component via supervised learning, which is typically more sample efficient. Secondly, the modular nature of the model allows us to swap components easily: we can train a masking model on simulated images *or* real images (cf. Section 5.1), allowing us to use the same action model for both simulated and real tasks without re-training. In an end-to-end model, a change to the input space would likely require re-training from scratch. Finally, the factorized model output is more interpretable; when an action is unsuccessful, we can inspect our model to localize the point of failure, be it a lack of comprehension or a mistake while acting. This may not be evident from the agent's action alone. Note that our implementation of this factorization is not without disadvantages: if the language contains information on how to act this data could be lost.

In summary, our contributions are:

1. We introduce a novel Transformer-based model for mapping language commands to masks over an action space. Our approach's modular nature enables us to seamlessly integrate an existing Q-learning framework proposed by Hundt et al. [9] for block manipulation tasks.
2. We show our model succeeds in a variety of simulated and real data settings with orders of magnitude less data than typically available to Transformers, furthering the potential use of Transformers in the low-data regimes commonly found in robotics.
3. We demonstrate this model's utility on an existing dataset (introduced by Bisk et al. [10]) involving a challenging block re-arrangement task with *crowdsourced* natural language commands containing syntactic diversity as well as complex natural language phenomena.
4. We release our real and simulated datasets, plus a filtered version of the Bisk et al. [10] dataset containing the subset of examples feasible under a realistic model of physics.

## 2   Related Work

We describe related work in the areas of natural language instruction following on robots, Q-Learning for multistep manipulation tasks, and Transformer models for Computer Vision.

**Robots and Natural Language Commands –** We follow the Problem-Solution Sequences (PSS) paradigm [11], where a multi-step manipulation task (e.g. "build a stack") is framed as a sequence of incremental images or state representations. These may be paired with per-step instructions on how to accomplish the intermediate goal (e.g., "stack the red block on the blue block"). Several PSS datasets have been proposed [11, 10] and modeled in 2D [4, 12] and 3D [10]. We translate the 3D dataset proposed by Bisk et al. [10] to a simulated environment in Section 5.2. The sequential nature of PSS relates to the hierarchical RL setting explored by Andreas et al. [13], who learn task-specific sub-policies indexed by linguistic sub-goals. Similar approaches are applied to zero-shot sub-goal sequence generalization[14], non-linearly-ordered sub-goal execution[15], and choosing interpretable linguistic sub-goals [16]. While we also assume access to step-wise annotated instructions, we focus on guiding a low-level robotic manipulation policy with a supervised masking model rather than a task-level policy. Gangopadhyay et al. [17] and Sun et al. [18] learn policies guided by symbolic programs, while Perera et al. [19] first parse language into a symbolic representations. Our masking-based method ties the language directly to the perceptual space, eschewing intermediate symbol ontologies. Language has been used for reward-shaping [20, 21, 22], and reward model learning [23]; we instead use the language-agnostic reward shaping in SPOT-Q. Stepputtis et al. [24] and Lynch and Sermanet [25] use language-conditioned imitation learning for manipulation policies; we instead use Q-learning. TransporterNets [26] learn to perform multi-step tasks from demonstration, but lacks language input. Like our work, Concept2Robot [27] completes language-described tasks by combining learning from demonstration and RL. However, it differs from our system along multiple axes: we examine challenging multi-step tasks with *progress reversal*[9]; we succeed on much smaller datasets; we consider crowdsourced commands; and we evaluate on real robot data. Nguyen et al. [28] guide object retrieval with language, but do not model placement.

Visual Goal Prediction (VGP), introduced by Misra et al. [4] for language-conditioned navigation [29, 30, 4, 8, 7, 5], frames the problem of choosing an action region as a semantic segmentation task in which each pixel as a separate binary decision: act or do not act. VGP has directed a simulated [5] and real quadcopter [8, 7]. We implement VGP with UNet[31, 4, 5, 7, 8], as a baseline.

**Q-learning for Multi-Step Tasks –** Mohammed et al. [32] reviews RL for robot grasping. Zeng et al. [33] shows Q-Learning outperforms supervised action segmentation for table decluttering[34, 35, 36], instead, "Good Robot!" [9] segments actions worth exploring to accelerate Q-Learning of single-task, multi-step robot learning, all without language input. "Good Robot!" [9] was also the first to complete multi-step tasks with consideration of *progress reversal*, a failure case where actions undo previous progress such that $\Omega(n)$ additional actions are required to recover, *e.g.* toppling the tower in a partially completed stacking task. We segment actions worth exploring via linguistic input for multi-step, multi-task execution considering *progress reversal* and low-data[3, 34] conditions.

**Transformer Encoders for Vision –** Though Transformer encoders, which build representations using self-attention, have become standard in NLP, computer vision tasks are still dominated by convolutional models, of which UNet [31] is a paradigmatic instance. The use of Transformers in vision tasks has been held back by the quadratic complexity of the Transformer's self-attention layer; an image input treated as a sequence of pixels would quickly become computationally intractable [37]. Dosovitskiy et al. [37] introduce a minimal modification to perform image recognition using Transformers, tiling the image into square patches, which they concatenate into a sequence and augment with learned linear positional embeddings. Though the model presented by Dosovitskiy et al. performs image-level recognition well, it cannot perform pixel-level tasks (e.g., semantic segmentation) as-is. Ranftl et al. [38] present an extension to this model for pixel-level prediction.

## 3 Models

We propose a Transformer-based model to convert image and natural language inputs into separate masks for grasping and placing, thereby allowing users to guide multi-step manipulation tasks.

We begin by tiling the $n \times n$ birds-eye image of the scene with $C$ colored blocks into $k \times k$ patches, whose RGB pixel values are concatenated into one vector, resulting in patch vectors $I = i_1, \ldots, i_L$. The resulting vectors are augmented with a fixed sinusoidal positional encoding [39]. Similarly, each token in the command string $W = w_1, \ldots w_m$ is embedded into a continuous space resulting in vectors $e_1, \ldots, e_m$, and those vectors are augmented with positional encodings as well.

The goal is to train a model that produces a pair of language-defined spatial masks $\hat{M}^{\text{place}} \in \mathbb{R}^{n \times n}$ and $\hat{M}^{\text{grasp}} \in \mathbb{R}^{n \times n}$, and an optional pair of state reconstructions $\hat{S}^{\text{grasp}}, \hat{S}^{\text{place}} \in \mathbb{R}^{C \times n \times n}$ given by:

$$\hat{M}^t = R\Big(\sigma\big(H_m^T f_t\big(f_{\text{shared}}([I;W])_{0:k}\big)\big)\Big), \ \hat{S}^t = R\Big(\sigma\big(H_s^T f_t\big(f_{\text{shared}}([I;W])_{0:k}\big)\big)\Big) \qquad (1)$$

where $f_\cdot$ are parameterized by Transformer encoders with output size $k$, $H_m \in \mathbb{R}^{k \times 2}$, $H_s \in \mathbb{R}^{k \times C}$, $\sigma$ is the softmax function, and $t \in \{\text{grasp, place}\}$. The reconstruction function $R$ up-samples $k$ patches by copying the patch value across each of its constituent pixels, resulting in an $n \times n$ image. Note that structurally, the "shared," "grasp," and "place" Transformers are identical, but while the "shared" Transformer is used both for grasp and place prediction, the others are task-specific.

We train the masking model separately from the manipulation model. For the latter model, we use a pre-trained SPOT-Q model from Hundt et al. [9], which produces pixel-wise Q-values for picking and placing, making it easy to integrate with our approach. Furthermore, pixel-wise Q-values facilitate potential future work on extending our method to non-cubic shapes. All masking models are trained with a multi-task loss, composed of a pixel-wise binary cross-entropy loss for the masks, and an optional auxiliary reconstruction loss:

$$\mathcal{L}_t = -\frac{1}{N^2}\Bigg(\sum_{i,j} M_{i,j}^t \log(\hat{M}_{i,j}^t) + \lambda \sum_{i,j} S_{i,j}^t \log(\hat{S}_{i,j}^t)\Bigg), \ \mathcal{L} = \mathcal{L}_{grasp} + \mathcal{L}_{place} \qquad (2)$$

The reconstruction loss here forces the model to semantically segment the image by identifying pixel with its block ID, or labeling it as background. As in Nguyen and Salazar [40], we use a pre-norm Transformer layer with scaled initialization. Our learning rate schedule is based on Vaswani et al. [39], with a varied number of warmup steps as in Nguyen and Salazar. A comprehensive report of all hyperparameters and a figure of our architecture is given in Appendix A.

**Q-Learning –** Our model relies on a learned function from states and actions to expected rewards, $Q(s_t, a)$, which we learn via the SPOT-Q algorithm [9], which introduces novel reward shaping processes for improving efficiency. More details are given in Appendix A.3. Given a pixel-wise $n \times n$ state space of an image and two actions, grasp and place, the output of this function can be expressed as a tensor $Q \in \mathbb{R}^{n \times n \times 2}$.

## 4 Datasets and Resources

### 4.1 Simulation Experiments with Synthetic Commands

We create a dataset of commands and grasp/place masks to train our masking model and demonstrate its performance in simulated environment on a robot arm. Using trajectories from Hundt et al. [9], we annotate successful grasp-and-place action pairs with synthetic commands generated from a template, and use simulation state information to construct the reference masks $M^t$ and state information $S^t$. While synthetic language makes simplifying assumptions about the syntactic and lexical diversity of natural language, it provides a controllable test-bed for symbolic reasoning, and is often used to study compositionality [41, 42, 43], as we do in Section 5.1. For row-making, the template is "move the `color_a` block next to the `color_b` block", and for stacking the template is "stack the `color_a` block on the `color_b` block". We train supervised VGP models (cf. Section 3) on these commands, and combine the produced masks with the Q-values from the SPOT-Q framework in a simulated environment to execute the actions in order. The visual input to our models is the concatenation of a birds-eye color image and depth image. The training data is augmented by flipping each image across 4 axes, manipulating the command accordingly, as well as by the addition of pixel-wise Gaussian noise drawn from $\mathcal{N}(0, 0.05)$. This results in train/dev/test splits of $5520/46/46$ and $8970/74/76$ for row-making and stacking, respectively, where each sample is an input image and command paired with a reference place and grasp mask. Experiments and results on VGP metrics and in the simulation for this data are given in Section 5.1.

### 4.2 Transfer to Real Images

In order to be useful in real-life settings, our method must also be applicable to real data. However, unlike simulated data, data from a real robot is expensive and time-consuming to collect. Therefore, it is natural to ask: how much real data is needed to train an adequate VGP model?

The Hundt et al. [9] real robot trial logs include 1496 actions each for row-making and stacking. We split these into an 80/10/10 train/dev/test split and augment the examples by flipping, producing $5980, 747, 748$ datapoints. Image collection details are given in Appendix B.2. Given the small size of the real dataset, we pre-train a model on simulated images and fine-tune it with the real images.

## 4.3 Compositionality

In linguistics, compositionality is typically taken to mean that the meaning of a whole utterance is *composed* of the meanings of its parts; the meaning of "stack the red block on the blue block" is in part made up of the independent meanings of "red" and "blue". Compositionality lets speakers re-combine known symbols in new ways, underpinning human language use. In our task, a compositional reasoner should correctly execute "stack the red block on the blue block" (referred to here as `stack(r, b)`) without having seen that combination; it should be able to localize red and blue blocks from other color contexts (e.g. `stack(r, g)`, `stack(y, b)`, etc.). However, there is no explicit constraint during training to enforce this; the model could learn to interpret each of the 23 possible color pairs atomically. We test our model's compositional reasoning via its ability to generalize to unseen color combinations. We hold out 6 color pairs during training (26% of the 23 order-invariant pairs) and test exclusively on those pairs. For example, we train a masking model with all color combinations except those made only of red, blue, yellow, and green blocks and then test only on `stack({r,b,g,y},{r,b,g,y})` examples. We hold-out each subset, training and evaluating a model as described above, average the metrics across all subsets, and compare against a random train and test split baseline, where all color pairs are seen at train time. For efficiency, we train on each task separately and do not augment with Gaussian noise.

## 4.4 Naturalistic Language Commands with Complex Scenes

Language sourced from human speakers is syntactically and lexically diverse, with mistakes, misspellings, and other idiosyncrasies that make far more challenging to model than template data. Analyzing synthetic language showcases our model's ability to integrate discrete symbolic commands with visual perception and action, as well as our model's ability to generalize in a compositional fashion, but this does not imply that our model generalizes to the complexities of real language.

We examine a 3D block manipulation dataset with human natural language instructions that was introduced by Bisk et al. [10] (cf. Fig. 2). This dataset is set apart from other block-world data by its focus on complex instructions. Rather than using synthetic utterances, instructions were sourced from crowdworkers (9 per example), who were shown a pair of images and asked to provide instructions to a robot, with the goal of manipulating the blocks in the first image such that they result in the second image. This collection method result in more syntactically diverse utterances with complex linguistic concepts, such as "mirroring" and "balancing".

The dataset is comprised of a sequence of state representations paired with natural language instructions on how to transition between states. While the blocks dataset is designed for robot manipulation, several assumptions were made to simplify annotation by crowdworkers. Crucially, physics in the environment were turned off while the structures were being constructed and annotated, resulting in some

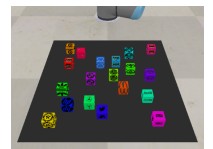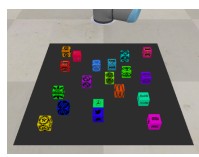

"Mercedes Benz will move right until it is above twitter"

Figure 2: Bisk et al. [10] blocks data sample.

physically infeasible structures. To facilitate research in the blocks domain, and make the dataset more useful for training and evaluating embodied agents, we create a filtered version of the dataset, where physically infeasible examples are removed. Our filtering process removes $49.34\%, 43.93\%$ and $38.23\%$ of train, dev, and test, respectively. Appendix C.1 contains more details on the translation process and data preprocessing. To model the dataset's world state sequences in the VGP framework, we translate the 3D Cartesian coordinates of each block given by the dataset's state representation into a top-down grid view, where each pixel corresponds to the ID of the block located at that position. We refer to this image as the scene's "state representation."

While the blocks dataset contains all the state information required to produce image-like state representations, this cannot generally be assumed to be the case. A more likely scenario mimicks Section 4.1, where the model's input is a combined color and depth image. After translating the dataset into the simulated environment used in Section 4.1, we collect color and depth images for the whole blocks dataset. In the original dataset, each block type is identified by a company logo. Since the logos were downsampled by the simulator, we also assign a color to each type. The demographics and labor practices underlying its original creation are unknown [10, 44]. We filtered their dataset without crowdworkers; and it is intended for robotics research, not product deployment. Experiments and results for this dataset are in Sec. 5.2.

# 5 Experiments

## 5.1 Simulation Experiments with Synthetic Commands

An operator guiding a robot through a manipulation task may have a particular outcome in mind: for example, a stack with a particular order of blocks. Language can naturally express such constraints; i.e. "stack the red block on the yellow block." Following the template generation and data annotation in Section 4.1, we train baseline and Transformer VGP models to produce output masks $\hat{M}^{\text{grasp}}$ and $\hat{M}^{\text{place}}$ and combine them with a SPOT-Q model to manipulate blocks in a simulated environment. These masks are filtered and thresholded using 0-1 block masks, following Hundt et al. [9]. To evaluate the VGP system in isolation, we report the following metrics:

- *Block Accuracy* is the percentage predictions for which the grasp mask overlaps with the correct block, i.e. the model chooses the correct block to pick up. Higher is better.
- *Idealized Error Score (IES)* captures the performance of an idealized robot on the manipulation task, choosing a block to move using the grasp mask, and a location using the place mask. After the chosen block is teleported to the predicted location, we measure the Euclidean distance between the block's new location and its reference location in block lengths. Lower is better.

Our baseline is a UNet-based VGP model; specifically, we extend the LINGUNET model presented by Misra et al. [4] with an attentional fusion mechanism. While the original LINGUNET used the final state of an LSTM for the language representation, we use a weighted average of BiLSTM states produced by an attention mechanism. We also ran a end-to-end Q-Learning experiment that backpropagates through both the FCN and Transformer, but the model did not converge.

Further details on the masking, as well as the simulation environment, can be found in Appendix B.1. The filtered/thresholded masks $\hat{M}^{\text{grasp}}$ and $\hat{M}^{\text{place}}$ are stacked into a single mask $\hat{M} \in \mathbb{R}^{n \times n \times 2}$. A final matrix of $Q'$ of Q values constrained by the language is given by $Q' = \hat{M} \odot Q$, where $\odot$ is the Hadamard product; $Q'$ is then used to choose an action and a location at which to grasp/place in the simulation.

We count stacks and rows as successful only with the specified color order, also tallying the number of actions this required and the percentage of times the gripper picked up the correct block. We contrast our model with the SPOT-Q framework alone, which has no access to the order constraints, and with our UNet baseline. Each model was evaluated with the same SPOT-Q model and tested on 100 trials. Hundt et al. [9] set a 30 action limit for failed trials; we added several evaluation heuristics to terminate failed trials when an irrecoverable state was reached, measuring color ordering errors. Stacking tasks use 8 colors and row-making tasks use 4.

Table 1 shows the visual goal prediction metrics for all model variants trained on one or both tasks. We see that the Transformer-based models (TFMR) uniformly out-perform the UNet baseline across all metrics, often by large margins. For both UNet and Transformer architectures, training on two tasks simultaneously covers a broader range of semantically different examples with a corresponding increase in performance on all metrics. Notably, these examples differ by task: firstly, row-making uses a subset of the colors available in stacking. Secondly, while the grasp data across both tasks is roughly identical, the data for place masks is quite different. For example, both "stack the blue block on the red block" and "move the blue block next to the red block" result in grasp masks over the blue block, but whereas the place mask for stacking picks out the block on which to stack (the red block), the row mask specifies a location *near* the target block. Thus, it is noteworthy that combining tasks benefits not only the block accuracy, but also the IES.

Table 1: Visual goal prediction (VGP) metrics for models trained on one or both tasks. Higher accuracy and lower Idealized Error Score (IES, measured in blocks), is better. TFMR is Transformer.

| Model | Train | Test | Block Acc. | IES |
|---|---|---|---|---|
| UNet | Rows | Rows | 57.1% | 2.4 |
| TFMR | Rows | Rows | 84.8% | 0.9 |
| UNet | Both | Rows | 66.5% | 1.2 |
| TFMR | Both | Rows | **91.3%** | **0.6** |
| UNet | Stacks | Stacks | 27.9% | 1.0 |
| TFMR | Stacks | Stacks | 81.5% | **0.4** |
| UNet | Both | Stacks | 75.2% | 0.6 |
| TFMR | Both | Stacks | **90.4%** | **0.4** |

Table 2: Simulation Metrics. Higher task and color grasping success percentages are better. A lower average number of actions per trial is better.

| Task | Model | Task % | Color % | Actions |
|---|---|---|---|---|
| Rows | RL only | 10% | 31.2% | 15.3 |
| Rows | UNet | 10% | 32.5% | 19.2 |
| Rows | TFMR | **53%** | **76.7%** | **12.3** |
| Stacks | RL only | 0% | 13.0% | NaN |
| Stacks | UNet | 0% | 15.2% | NaN |
| Stacks | TFMR | **29%** | **78.6%** | **11.6** |

Table 2 shows the task completion percentage for the best UNet and Transformer models (trained on both tasks) in the simulated environment, as well as the percentage of successful grasps that picked up the correct color, and the average number of actions taken to complete a successful trial. In the aggregate, we see that high performance on VGP metrics does not translate to equivalent performance in the simulation environment. For example, both for stacking and row-making, the percentage of correct color grasps is substantially lower that the VGP equivalent metric, block accuracy. For the Transformer model, this may be due to the imprecision of the patch-wise tiling of the image: when heuristically combined with a common-sense mask, a patch may cover multiple adjacent blocks. This is an inherent limitation of our approach. When compared to the RL-only and UNet baselines, our Transformer-based model succeeds in significantly more trials. Additionally, the mean number of actions across successful trials is lower, indicating that the Transformer-based masks result in fewer actions per successful trial. Note that for stacking, both baselines succeeded on 0 out of the 100 trials; due to toppling and occlusion, stacking poses a significantly harder task than row-making, and it is harder to recover from a misplaced block, even if the trial is not ended. We see that even a fairly high block accuracy and low IES are insufficient for high task performance. This highlights the one of the main challenges of multistep tasks: small error rates compound across multiple actions, yielding low task performance. A full error analysis is given in Appendix B.3

**Transfer to Real Images –** In Figure 3, we plot the IES of row-making and stacking models when trained on increasing subsets of real data. We see that for both rows and stacks, performance improves slightly as we increase the amount of training data, with diminishing returns after 30% of the available data (1794 examples) are used. Given that Transformers typically require several orders of magnitude more data to reach good performance [45, 40, 46], the fact that our architecture can be trained with mere thousands of simulated training examples, and

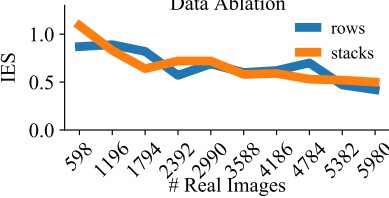

Figure 3: IES as a function of the % of real fine-tuning data used. IES is error measured in blocks, lower is better.

successfully finetuned with even fewer real examples, extracted from only a few hundred real images, represents a significant achievement. For a proof-of-concept demonstration of our model running on a real robot, see Appendix B.4 and the supplementary materials.

**Compositionality –** By generating held-out data splits as described in Section 4.3, we can test the ability of our model to generalize compositionally, i.e. to recombine known symbols in new contexts. A model with compositional color representations will be able to combine colors across contexts; for example, if it has seen red blocks in the context of green blocks, it should generalize to a blue block context. We would therefore expect the model to perform roughly on par with a model trained on a i.i.d. data split, where all contexts are seen at train time. Conversely, if the model has learned to interpret each utterance atomically, we should expect much lower held-out performance.

Our results are reported in Table 3. Both rows and stacks see a slight drop in block accuracy from the baseline. Nevertheless, the lack of a large gap between the baseline and the held-out conditions suggests that the model has largely learned to generalize compositionally to unseen block combinations. This suggests that, although the model certainly has the capacity to memorize 23 combinations, and although it is not trained to perform compositional reasoning, it learns to bind symbols to percepts in

Table 3: VGP metrics on a held-out subset of unseen color pairs. Baseline models trained on a randomly-shuffled split.

| Task | Data Split | Block Acc. | IES |
|------|-----------|-----------|-----|
| Stacks | Random | 85.7% | 0.4 |
| Rows | Random | 89.1% | 1.7 |
| Stacks | Unseen | 84.8 % | 0.6 |
| Rows | Unseen | 84.3% | 1.3 |

a way that generalizes across contexts (i.e. "red block" refers to one particular block, independent of its context). These results are promising but limited to simulated language; accordingly, in Section 5.2 we evaluate our model with real language data.

## 5.2 Naturalistic Language Commands with Complex Scenes

In this experiment we model the naturalistic data described in Section 4.4, where state representations are coupled instead with diverse crowdsourced commands, in contrast to the template generated commands in Section 5.1. We train models on state representations (cf. Section 4.4) as well as birds-eye color and depth images. This highlights the flexibility of our model: we are able to train from images and from state representations with no modifications to the architecture.

Large pre-trained encoders have been shown to encode useful linguistic features at a number of levels of analysis [47, 48, 49]; accordingly, we make use of `bert-base-cased` [50] as an input feature, contrasting it with GloVe [51] word embeddings. Tuning and metrics are as in Sec. 5.1.

Table 4 shows the VGP metrics for models trained from image-like representations of the state (described in Section 4.4) as well as from images. For brevity, only the best model for each embedding type is reported, with full results in Appendix C.5. On state representations, the Transformer-based models show a clear advantage over the baseline, with IES being substantially

Table 4: VGP metrics show that best Transformer models out-perform UNet, both on state and image inputs.

| Model | Input | $k$ | Block Acc. | IES |
|---|---|---|---|---|
| UNet + GloVe | State | 1 | 56.5% | 3.2 |
| UNet + GloVe | Image | 1 | 63.4% | 3.1 |
| TFMR + GloVe | State | 2 | 88.4% | **2.1** |
| TFMR + GloVe | Image | 2 | *88.7%* | 2.8 |
| TFMR + BERT | State | 4 | **90.5%** | **2.1** |
| TFMR + BERT | Image | 2 | 83.6 | 3.4 |

lower for both Transformer models. When using GloVe embeddings with state representation input, $2 \times 2$ patches resulted in the best performance; a patch size of 4 yielded the best results for BERT embeddings. Overall, while the BERT-based model has slightly higher block accuracy, the IES are equal between the two models when trained on state representations. While these metrics measure per-step performance, these results suggest that our masking model can be applied to real as well as simulated language.

In Table 4, we also see that the performance shifts substantially when changing the input from state to images. While the baseline improves slightly in IES, the Transformers' performance decreases dramatically. These results stand in contrast to those in Section 5.1, where a Transformer trained on images unequivocally outperformed the baseline. Nevertheless, the best Transformer model still outperforms the baseline. We speculate that the baseline models may perform better here because the architecture was developed for image segmentation problems. Interestingly, while a BERT-based Transformer performed well when trained from state, the same models' performance here is greatly diminished. Improving the Transformer's real-world performance is a direction for future work.

**Data Translation and Filtering** In Section 5.1, we observed that high performance in a static evaluation against a dataset did not entail equally high performance when using a realistic robot for multi-step tasks in a simulated environment. In its current form, the blocks dataset can only be evaluated using static metrics. Furthermore, the images provided in the dataset are taken from a fixed 45 degree projection, and our method assumes access to birds-eye images. To obtain birds-eye images of the environment and to facilitate further research into combining rich, naturalistic language with action in manipulation tasks, we translate the dataset into the same environment used in Section 5.1, and filter the dataset to only physically feasible examples; this process is described in Section 4.4.

Table 5 reports the VGP metrics on the the physically feasible subset of the data for the Transformer models. Firstly, we see that performance decreases across most metrics. However, for UNet, IES improves on the physically feasible subset. In contrast, the Transformer models decrease in performance across both metrics. Nevertheless, the Transformer model still performs on par

Table 5: VGP metrics for models trained on images of the physically feasible subset of the blocks dataset. * indicates significant improvement over UNet as per the Wilcoxon ranked-sign test.

| Model | Input | $k$ | Block Acc. | IES |
|---|---|---|---|---|
| UNet + GloVe | Image | 1 | 56.4 | **3.0** |
| TFMR + GloVe | Image | 2 | **84.4**\* | **3.0** |
| TFMR + BERT | Image | 2 | 78.9\* | 3.6 |

with the baseline on IES, and substantially better on block accuracy. These results suggest that the physically feasible subset is more challenging to model, even when evaluating the models statically. In other words, even when the evaluation and models are physically unconstrained, modeling only physically feasible scenes may be harder than modeling all scenes. This result highlights the difficulty of developing embodied agents and the importance of testing them in realistic environments.

## 6   Conclusion

We have presented a general approach to multi-step rearrangement and manipulation tasks that factorizes each step into learning how to act and learning where to act. To decide where to act, we have presented a Transformer-based model for converting state representations and natural language instructions to masks over an action space that learns with remarkably few examples. We have shown the ability of our model to act in a realistic simulated environment for two multi-step block rearrangement tasks, and naturally learn a compositional mapping from text to actions in a perceptual space. We have demonstrated that our model performs well under both these realistic physical assumptions and with real and complex natural language via a dataset of crowdsourced instructions.

# 7  Acknowledgements

We thank the anonymous reviewers and the area chair for their input and engagement. We would also like to thank Benjamin Van Durme, Tae Soo Kim, and Jonathan Jones for their helpful feedback. This material is based upon work supported by NSF Award #1763705 and by the Office of Naval Research under grant N00014-19-1-2076. Elias Stengel-Eskin is supported by an NSF GRFP.

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

# A Model

## A.1 Transformer

Figure 4 shows a schematic of our Transformer-based VGP model, run on real robot data. The inputs are a bird's eye color image (and depth image) and a language command. The image is tiled into an array of non-overlapping square patches, which are flattened into a one-dimensional list by concatenating each row of tiles. The tokens of the language command are then concatenated to the end of that list. This concatenated input is passed through a shared set of Transformer encoder layers, which encodes the dependencies between words and the image. The output of the shared Transformer is then passed to two identical Transformer encoders with separate parameters, one for predicting a grasp mask, and one for predicting a place mask. This bifurcation is based on the observation that there are commonalities between producing the two types of masks (e.g. localizing colors, delineating blocks) but that the final tasks of producing grasp and place masks are distinct.

## A.2 Hyperparameters

The number of shared and combined layers is treated as a hyperparameter $n \in [0, 2, 4, 6]$, as well as the number of warmup steps $w \in [100, 400, 1000, 4000]$, the dropout probability $p \in [0.25, 0.33, 0.40]$, and the loss weight ratio $\gamma \in [0.01, 0.1, 0.2]$. Each model has a hidden dimension of 256 and a feed-forward dimension of 512. Due to the larger number of hyperparameters, we sample 70 random parameter configurations.

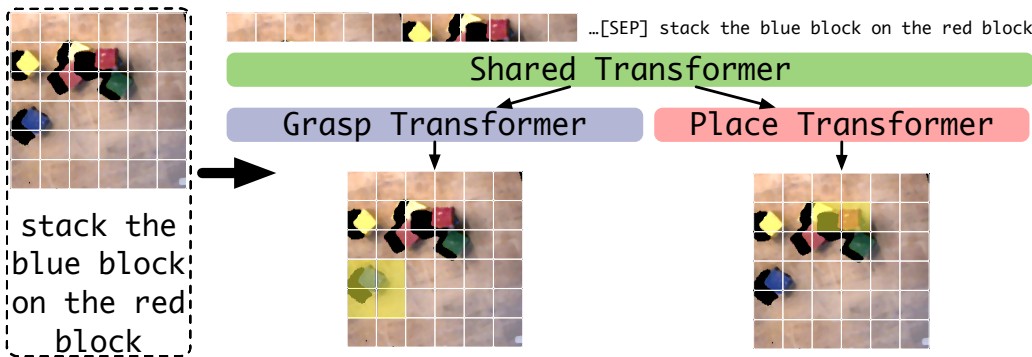

Figure 4: Our Transformer-based Visual Goal Prediction model, as run on real robot data. It takes an image and an instruction as input, then produces separate masks for grasping and placing.

## A.3 Q-Learning

Our model relies on a learned function from states and actions to expected rewards, $Q(s_t, a)$. This function is learned in the context of a Markov Decision Process $(S, A, P, R, \gamma)$ composed of a set of states $S$, a set of actions $A$, a transition function $P : S \times S \times A \to \mathbb{R}$, a reward function $R : S \times A \to \mathbb{R}$, and a discount factor $\gamma$, where $0 \leq \gamma \leq 1$. At time $t$, an agent observes state $S_t$ and chooses action $a_t = \pi(s_t)$, where $\pi$ is a policy. The function $Q$, approximating the expected reward for each state-action pair, can be used to extract a deterministic policy $\pi$ that maximizes the expected reward: $\pi(s_t) = \operatorname{argmax} Q(s_t, a), \forall a \in A$. We learn $Q$ to maximize the reward $R$ over time by minimizing $|Q(s_t, a_t) - y_t|$, where $y_t = R(s_{t+1}, a_t) + \gamma Q(s_{t+1}, \pi(s_{t+1}))$. In a multi-step task, the definition of $R$ has a major impact on the learning efficiency of the agent. Specifically, relying on a single reward at the end of a task yields a very sparse reward signal and results in inefficient learning. In the multi-step tasks considered here (stacking and row-making) there are intermediate subtasks that naturally lend themselves to reward shaping, i.e. providing smaller intermediate rewards to the agent when the chosen action reflects an incremental progression towards the goal state. The SPOT-Q algorithm, introduced by Hundt et al. [9], uses intermediate rewards to shape the learning of $Q(s_t, a)$ for row-making and stacking. We use SPOT-Q to learn the function $Q(s_t, a)$. Given a

pixel-wise $n \times n$ state space of an image and two actions (grasp and place), this function can be expressed as a tensor $Q \in \mathbb{R}^{n \times n \times 2}$.

## B  Experiment 1

### B.1  Combination Heuristics

In the case of stacking, following Hundt et al. [9], we intersect each mask with a common-sense mask that assigns a value of 0 where there are no blocks (i.e. an empty space cannot be the location of a grasp or a place). For row-making, we allow placing in empty locations, but intersect the grasp mask with a common-sense mask. For place and grasp actions in stacking, and grasp actions in row-making, we then find the maximum location $s^*$ in mask $M$: $s^* = \arg\max M$. Because the reconstruction function copies a patch value across all of its pixels, for a patch size of $p$ there will be $p^2$ pixels with the value of $s^*$. We intersect this patch with the common-sense mask to find the single block with the highest value in $M$. For place actions in row-making, we simply threshold $M$ to obtain a set of valid patches to intersect with the Q values. Both processes produce binary masks.

### B.2  Simulation

We use the CopelliaSim simulator [52] for experiments involving a simulated robot.

**Images**  All real images were collected via a UR-5 robot, Robotiq 2f-85 2-finger gripper, and Prime-sense Carmine RGB-D camera. All of these conditions are different from the simulator, except the UR5. The images were collected under varied lighting conditions. The data is saved in the same file format as the experiments in Tables 1 and 2. with a birds-eye view, $224 \times 224$ RBG and depth-map images, etc.

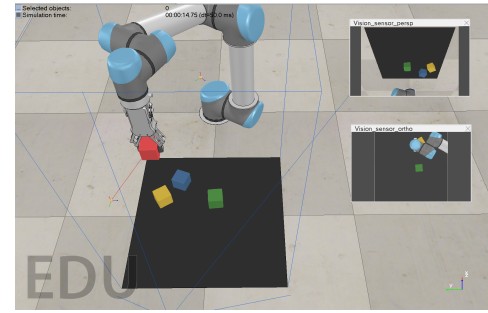

Figure 5: Simulation Environment

**Early Termination**  The step-by-step nature of the instructions means that the agent is sometimes unable to reverse course: for example, if the first instruction were "stack the green block on the blue block" but the agent mistakenly grasped the red block and placed it on the green block, the green block would become unreachable. Since neither the Q-value model nor the language understanding model were trained on unstacking tasks, there would be no way to reverse course here without adding an external observer. Thus, in these cases, we terminate the trial. Similarly, if the stack is toppled, we terminate the trial, as without an external observer we cannot determine which step of the process we have regressed to, as the partial toppling of stacks is not unusual. Because blocks are typically not occluded in row-making, we do not need these heuristics, and the only way for a trial to fail is by timing out after 30 actions.

**Simulator**  For experiments involving a simulated robot, the CopelliaSim simulator [52] was used. The simulated agent collects observations via a fixed RGB-D camera, whose images are project to a birds-eye view. The agent operates over a discretized spatial and angular action space, and movement to a particular location and angle is performed by an inverse kinematics solver built into the simulator.

### B.3  Error analysis

As mentioned in §5.1, stacking tasks may terminate early for a number of reasons, including irreversible actions that make a successful stack impossible. We conduct an error analysis to determine what percentage of the failures are due to errors in the perception/reasoning component (i.e. the masking) and what percentage are due to physical failures of the grasping arm. We find that 61.97% of the failures could be attributed to timing out after 30 actions. Qualitatively, we found that this

often happens when a block tumbles outside of the work area, rendering it impossible to grasp. Another 7.04% can be attributed to toppled stacks, where a stack height of 2 or 3 was reached before a place action destroyed the whole stack, ending the trial. Finally, 30.99% were due to incorrect block orderings which occluded crucial blocks; this is due to the 21.42% of grasp actions that picked up the incorrect block.

### B.4  Proof-of-Concept

Due to restrictions induced by the ongoing COVID-19 crisis, we were unable to run sufficient trials to include quantitative real results. Nevertheless, we include a proof-of-concept demonstration video in which we run the Transformer-based masking model with a real robot arm to successfully complete a real stack in the bottom to top order green, yellow, blue, red.

## C  Experiment 2

### C.1  Data Translation and Preprocessing

Note that in the original dataset, 4.82% of the examples have multiple blocks moving in a single frame. As this breaks the assumptions made in § 1, we ignore these examples both while training and evaluating.

For training, we convert the $64 \times 64$ state image grids into binary masks, where all elements are zero except the pixels corresponding to the block which is moved.

### C.2  Data Filtering

We filter out physically infeasible examples, which are typically due to unstable structures toppling or blocks being placed in overlapping areas. Because the images captured from the simulated environment are of a lower resolution than the originals, many of these logos are difficult to read in the simulated images; to aid with block discrimination, we assign a color to each block in addition to its logo. Table 6 reports the same statistics given in Bisk et al. [10] on the filtered subset of the data.

| Dataset | Configs | Types | Tokens | Utterances | Mean Length |
|---|---|---|---|---|---|
| Bisk et al. [10] | 100 | 1,820 | 233,544 | 12,975 | 18.0 |
| Physically Feasible | 100 | 1,434 | 133,083 | 7,398 | 18.0 |

Table 6: Dataset Statistics for feasible subset of the Bisk et al. [10] data

### C.3  Dataset Ambiguity

A remaining limitation of the dataset is that some descriptions are ambiguous and not reliably actionable, since annotators did not attempt to execute the described actions. Some of the natural language descriptions are ambiguous, as depicted in Fig. 7.

### C.4  Qualitative Error Analysis

Based on our results in Tables 4, we qualitatively examine some of the errors made by the Transformer-based model, where we observe several patterns. Looking at the validation examples with the highest IES, we observe that they often correspond to instances where the source block was mis-identified, leading the wrong block to be moved, and yielding a large IES. More interestingly, in the cases where the correct block was moved, the instruction often contains higher-order concepts as well as linguistic complexities such as ellipsis. For example, in Figure 6 there is a reference to a "disconnected square", which is a rare, higher-order geometric concept. In addition, the subsequent clause lacks an explicit reference back to the square; the annotator chose to leave this reference as implicit, given that the square is raised to a salient position by the previous clause. This type of noun-phrase ellipsis is common in natural language [53], and reflects the type of advanced pragmatic reasoning required to handle natural language.

Other examples of a higher-order concept instruction that the model performs poorly on are: "Take the Mercedes Benz block on the Burger King block without hiding the right top edge" (hiding),

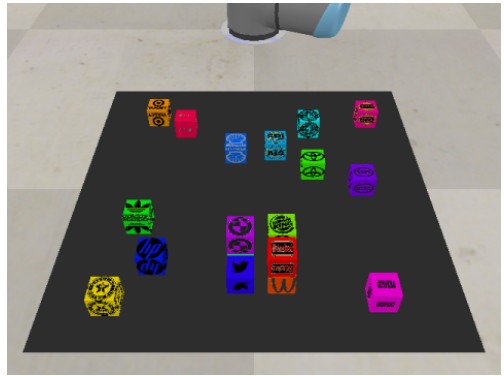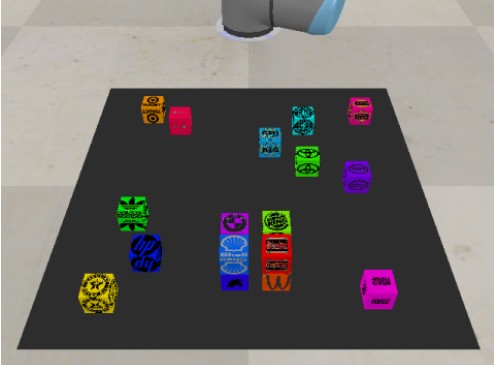

"There is a disconnected "square" at the bottom.
Place Shell on top of the lower left corner."

Figure 6: A particularly challenging example for the model. Not only does this involve a higher-order concept (disconnected square) but it also incorporates noun-phrase ellipsis.

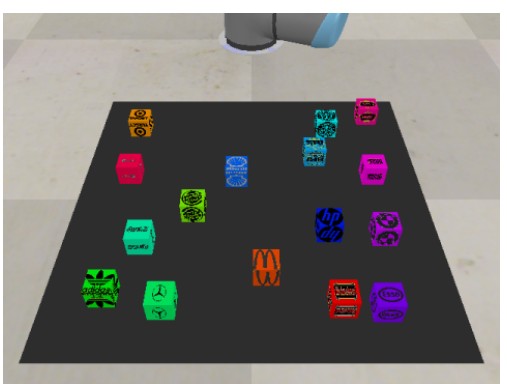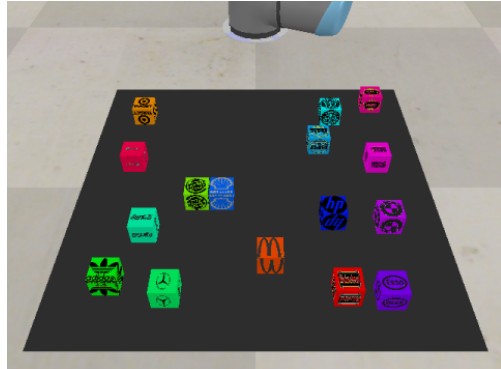

"Add Shell as the second block in the four-block row."

Figure 7: Ambiguous command makes reference to future steps: a row of 4 is later constructed, but at the time of construction, the agent has no access to this information.

"targt [sic] goes 1/2 under Adidas with the right side hanging off" (hanging), and "take the Stella Artois block and place it on top of the Nvidia block, lined up perfectly" (perfectly). The model also struggles with long coreference chains, even when the anaphora are explicit, e.g. "Place the block that is to the right of the Starbucks block and make it the highest block on the board by placing it on the Mercedes block. It should be in line with the bottom block."

## C.5 Full results for Experiment 2

Table 7: Full VGP results for the blocks dataset

| Model | Embedding | Recon. Loss | Input | $k$ | Block Acc. | IES |
|---|---|---|---|---|---|---|
| UNet | GloVe | Yes | State | 1 | 55.3 | 3.3 |
| UNet | GloVe | No | State | 1 | 56.5 | 3.2 |
| UNet | GloVe | Yes | Image | 1 | 53.6 | 3.1 |
| UNet | GloVe | No | Image | 1 | 63.4 | 3.1 |
| Transformer | GloVe | Yes | State | 4 | 90.7 | 2.2 |
| Transformer | GloVe | No | State | 4 | 92.8 | 2.3 |
| Transformer | GloVe | Yes | State | 2 | 90.8 | 2.3 |
| Transformer | GloVe | No | State | 2 | 88.4 | 2.1 |
| Transformer | BERT | Yes | State | 4 | 88.9 | 2.6 |
| Transformer | BERT | No | State | 4 | 90.5 | 2.1 |
| Transformer | BERT | Yes | State | 2 | 89.0 | 2.4 |
| Transformer | BERT | No | State | 2 | 78.7 | 2.8 |
| Transformer | GloVe | Yes | Image | 4 | 88.9 | 3.4 |
| Transformer | GloVe | No | Image | 4 | 85.5 | 3.6 |
| Transformer | GloVe | Yes | Image | 2 | 79.3 | 3.2 |
| Transformer | GloVe | No | Image | 2 | 88.7 | 2.8 |
| Transformer | BERT | Yes | Image | 4 | 89.5 | 3.5 |
| Transformer | BERT | No | Image | 4 | 72.1 | 3.8 |
| Transformer | BERT | Yes | Image | 2 | 90.1 | 3.3 |
| Transformer | BERT | No | Image | 2 | 83.7 | 3.1 |

Table 8: Full VGP results for the blocks dataset on the physically feasible subset.

| Model | Embedding | Recon. Loss | Input | $k$ | Block Acc. | IES |
|---|---|---|---|---|---|---|
| UNet | GloVe | Yes | State | 1 | 47.8 | 3.1 |
| UNet | GloVe | No | State | 1 | 49.1 | 3.1 |
| UNet | GloVe | Yes | Image | 1 | 45.5 | 3.0 |
| UNet | GloVe | No | Image | 1 | 56.4 | 3.0 |
| Transformer | GloVe | Yes | State | 4 | 88.2 | 2.4 |
| Transformer | GloVe | No | State | 4 | 90.3 | 2.4 |
| Transformer | GloVe | Yes | State | 2 | 89.1 | 2.6 |
| Transformer | GloVe | No | State | 2 | 85.1 | 2.4 |
| Transformer | BERT | Yes | State | 4 | 86.8 | 2.7 |
| Transformer | BERT | No | State | 4 | 88.3 | 2.2 |
| Transformer | BERT | Yes | State | 2 | 88.4 | 2.6 |
| Transformer | BERT | No | State | 2 | 72.0 | 3.0 |
| Transformer | GloVe | Yes | Image | 4 | 86.3 | 3.4 |
| Transformer | GloVe | No | Image | 4 | 83.0 | 3.7 |
| Transformer | GloVe | Yes | Image | 2 | 75.7 | 3.2 |
| Transformer | GloVe | No | Image | 2 | 84.4 | 3.0 |
| Transformer | BERT | Yes | Image | 4 | 88.3 | 3.7 |
| Transformer | BERT | No | Image | 4 | 66.2 | 3.7 |
| Transformer | BERT | Yes | Image | 2 | 88.5 | 3.4 |
| Transformer | BERT | No | Image | 2 | 86.3 | 3.5 |

