# OpenReview forum: "Guiding Multi-Step Rearrangement Tasks with Natural Language Instructions"
_robot-learning.org/CoRL/2021/Conference — CoRL2021 Poster_

### Official Review · Reviewer_kWqo · 2021-07-17

**Originality:** Fair
**Technical Quality:** Good
**Clarity Of Presentation:** Good
**Impact:** 3

**Recommendation:**

Weak Reject: I recommend rejecting the paper, but will not argue for my recommendation if the majority of other reviewers have a different opinion.

**Summary:**


This paper addresses the problem of learning to perform block rearrangement tasks described by natural language instructions. To this end, the paper proposes a framework that decouples the problem into two sub-problems: where to act and how to act. Specifically, it proposes a Transformer-based model to map images and language instructions to masks over grasp and or place locations. Then, a pre-trained policy performs actions according to its Q function that provides pixel-wise Q values and predicted masks. The experiments show the proposed framework can reliably solve the task. Ablation studies verify design choices such as language models (GloVe and BERT), the encoder architecture (UNet and Transformer), etc. I believe this work studies a promising research direction (i.e. language instruction guided agents) and proposes a reasonable framework to address the problem. Yet, I am concerned with the narrow scope of the problem and the limited novelty and technical contributions.

**Issues:**


Described in the strengths and weaknesses section.

**Reviewer Expertise:**

Good: General knowledge of the area

**Strengths And Weaknesses:**


## Paper strengths and contributions

**Motivation**
- Leveraging natural language descriptions to instruct robots is easy and natural especially for non-expert users and flexible.
- Employing natural language instructions for learning a multi-task policy allows leveraging semantic similarity across tasks, which potentially allows generalizing to unseen tasks.
- Decoupling learning where to act and how to act could potentially be more sample efficient compared to learning an RL policy from scratch purely from reward signals. Also, such as framework is more interpretable since we can tell if it fails to predict correct locations to apply actions or it just cannot precisely perform actions.

**Technical contributions**
- Predicting segmentation maps indicating where to grasp or place seems effective
- Employing a Transformer to process natural language instruction yields better performance compared to UNet.

**Ablation study**
Ablation studies justify some of the design choices:
- UNet vs. Transformer: the results show that employing a Transformer model yields better performance compared to an UNet baseline.
- GloVe vs. BERT: the BERT-based Transformer has slightly higher block accuracy and performs well when trained from states.

**Experimental results**
The experimental results show that:
- The proposed Transformer-based visual goal prediction (VGP) model can predict segmentation maps for grasping and placing reasonably well in terms of block accuracy and idealized error score (IES).
- The proposed framework achieves a higher task completion percentage with a lower average number of actions per trial.
- The proposed framework can learn from thousands of simulated training images and successfully be fine-tuned with around 2k real images to achieve reasonable performance (with an IES around 0.7).
- The proposed framework can deal with unseen color pairs.

## Paper weaknesses and questions

**Novelty**
The novelty, from both the views of theoretical analysis aspect and empirical contribution perspective, is not clear or not explicitly mentioned.  Therefore, I do not find enough novelty from any aspects while the efforts put in for setting up and evaluating the proposed system are highly appreciated. To me, this seems more like a system paper than a learning paper.

**Two-stage pipeline**
This paper proposes to decouple the block rearrangement problem into where to act and how to act. While I do recognize the advantage of this decoupling scheme, there are still some issues:
- Error propagation: errors produced in each stage can accumulate and lead to undesired behaviors.
- The intermediate representations: intermediate representations (such as segmentation maps in this case) need to be chosen beforehand and the chosen one may not be the best for the entire system. The justification suggesting that a segmentation map is a good representation for this task is missing from this paper. Some other choices, such as object detection boxes, could have been evaluated.

**Limited problem scope**
While the proposed framework and ideas are applicable to many problem families, the only task considered in this paper is block rearrangement. I encourage the authors to apply the proposed framework to other domains to increase the significance of this work.

**Related work**
It would make the related work section more comprehensive by including some prior works that explore using formal languages (i.e. programs) as a task representation, including
- Modular multitask reinforcement learning with policy sketches
- Zero-Shot Task Generalization with Multi-Task Deep Reinforcement Learning
- Programmable agents
- Program Synthesis Guided Reinforcement Learning
- Program Guided Agent
- Hierarchical Program-Triggered Reinforcement Learning Agents For Automated Driving
- Reinforcement Learning of Implicit and Explicit Control Flow in Instructions

**RL only baseline**
It is not fair to compare the proposed framework against the RL-only baseline this way since the RL-only baseline does not leverage the segmentation map supervision and does not learn from other tasks like "SPOT-Q". Some fairer comparisons could be: (1) learning the RL policy with an auxiliary task of predicting segmentation maps, (2) learning a multi-task RL policy to solve the block rearrangement tasks with multiple configurations.

**Unseen color pairs**
While the ability to generalize to unseen color pairs (e.g. stack the red block on the blue block) is evaluated in this paper, the evaluation setup is very limited. It limits the number of unseen color pairs to only 1 but all other 55 combinations are seen. I would say holding out at least 10-20% makes more sense.

**Grid downscaleing factor**
Instead of predicting segmentation maps with the same size as the input images, this work proposes to downscale it. However, the downscaling factor would depend on the size of the object of interest and therefore it is not flexible.

**Summary Of Recommendation:**


I believe this work studies a promising research direction (i.e. language instruction guided agents) and proposes a reasonable framework to address the problem. Yet, I am concerned with the narrow scope of the problem and the limited novelty and technical contributions.

---

> ### Author Response · Authors · 2021-08-26
> **Authors’ initial kWqo Review Response, Thank you.**
>
> Thank you for your review and comments, especially the point regarding compositionality, which we have drawn on to conduct additional experiments that have strengthened our analysis.
> Could you please examine our updated revision pdf, with key changes in blue? We would really appreciate it!
>
> ---
>
> > 1. “Novelty The novelty, from both the views of theoretical analysis aspect and empirical contribution perspective, is not clear or not explicitly mentioned. Therefore, I do not find enough novelty from any aspects while the efforts put in for setting up and evaluating the proposed system are highly appreciated. To me, this seems more like a system paper than a learning paper.”
>
> We would like to point to the contributions of our work indicated by other reviewers:
> - Novels model for language guided control/actions (reviewer h8PP)
> - Filtered Bisk dataset (reviewer 1smN)
> - Ability to reuse pre-trained models (reviewer gQ8o)
>
> In addition to these, we would like to note that this work, to our knowledge, is the first to examine the highly cited Vision Transformer (ViT)[f] methods in conjunction with language instructions for guided multi-step manipulation. Furthermore, two of the three datasets (real and simulated block manipulation with language instructions) are novel, and the third (the physically feasible subset of the Bisk data) represents a novel reframing of an existing dataset. Finally, translating the Bisk dataset into a simulated environment is itself a novel contribution.
>
> ---
>
> > 2. “Two-stage pipeline This paper proposes to decouple the block rearrangement problem into where to act and how to act. While I do recognize the advantage of this decoupling scheme, there are still some issues:”
> > “Error propagation: errors produced in each stage can accumulate and lead to undesired behaviors.”
>
> We wholeheartedly agree with this point, and address it in section 5.1. As we mention in the metareviewer response, we experimented with a fully end-to-end model, which failed to converge. If such a model worked, it would be a way of reducing error propagation.
>
> ---
>
> > 3. “The intermediate representations: intermediate representations (such as segmentation maps in this case) need to be chosen beforehand and the chosen one may not be the best for the entire system. The justification suggesting that a segmentation map is a good representation for this task is missing from this paper. Some other choices, such as object detection boxes, could have been evaluated.”
>
> Thank you for this point, our introduction now refers to the basic concept of masking regions containing objects that was explored in Hundt et. al.[9], and our related work now notes their experimental justification. Other work backing similar practices includes [a,b,c,d].
> The rectangular nature of bounding boxes would hamper their utility in any future work aiming to manipulate irregular, non-rectangular, or deformable objects, such as a small sack of objects, since the bounding box region would have lower precision than the segmentation map.
>
> ---
>
> > “Limited problem scope While the proposed framework and ideas are applicable to many problem families, the only task considered in this paper is block rearrangement. I encourage the authors to apply the proposed framework to other domains to increase the significance of this work.”
>
> The current scope of the work, i.e. manipulation experiments on real and simulated colored blocks with templatically generated language, and masking experiments on state and image representations of complex scenes with natural language, is informed jointly by related work in block manipulation, access to block-based datasets, and the constraints of our physical environment. It is our hope here to lay the groundwork for future work, which will explore expanding our method to additional settings, including a more diverse set of objects to manipulate.
>
> ---
>
> > 4. “Related work It would make the related work section more comprehensive by including some prior works that explore using formal languages (i.e. programs) as a task representation”
>
> Thank you for suggesting these references, we have added the two papers that we could determine have been published.
>
> ---
>
> > 5. “RL only baseline It is not fair to compare the proposed framework against the RL-only baseline this way since the RL-only baseline does not leverage the segmentation map supervision and does not learn from other tasks like SQTP. Some fairer comparisons could be: (1) learning the RL policy with an auxiliary task of predicting segmentation maps, (2) learning a multi-task RL policy to solve the block rearrangement tasks with multiple configurations.”
>
> To ensure we understand the suggestion correctly: is this in reference to the RL-only baseline in Table 2? Could you provide a link to SQTP -- we are not familiar with the acronym.
>
> ---
>
> (Response continued below)

---

> > ### Author Response · Authors · 2021-08-26
> > **Continued Response**
> >
> > > 6. “Unseen color pairs While the ability to generalize to unseen color pairs (e.g. stack the red block on the blue block) is evaluated in this paper, the evaluation setup is very limited. It limits the number of unseen color pairs to only 1 but all other 55 combinations are seen. I would say holding out at least 10-20% makes more sense.”
> >
> > Thank you very much for this suggestion. We have updated the experiments in section 5. Specifically, instead of holding out one color pair at a time, for stacking, we hold out 6 pairs at time. This represents roughly 26% of the possible 23 color pairings. The results still reflect the model’s ability to reason compositionally. For row-making, we kept the experiments as-is, since there are only 4 colors used there, meaning only 6 possible color combinations.
> >
> > ---
> >
> > > 7. “Grid downscaleing factor Instead of predicting segmentation maps with the same size as the input images, this work proposes to downscale it. However, the downscaling factor would depend on the size of the object of interest and therefore it is not flexible.”
> >
> > As you indicate, ViT does operate over image patches, assigning each patch a label. However, the size of the segmentation map is the same as the size of the image, and the patch size is a hyperparameter which can be tuned, as we show in Section 5.2 as well as Appendix C5. Furthermore, the results in Section 5.2 indicate that the best patch size is often very small (2x2 pixels). While this does down-scale the image slightly, when combined with pixel-wise Q-values the result is often sufficiently precise for manipulation (cf. Section 5.1), and this is consistent with other approaches including the FCN in the RL model[9] underlying our method.
> >
> > Thanks again for your detailed response and for reviewing our submission.
> >
> > [a] Misra, Dipendra, John Langford, and Yoav Artzi. "Mapping Instructions and Visual Observations to Actions with Reinforcement Learning." Proceedings of the 2017 Conference on Empirical Methods in Natural Language Processing. 2017.
> > [b] .  Blukis, Valts, et al. "Following high-level navigation instructions on a simulated quadcopter with imitation learning." arXiv preprint arXiv:1806.00047 (2018).
> > [c] Blukis, Valts, Ross A. Knepper, and Yoav Artzi. "Few-shot Object Grounding and Mapping for Natural Language Robot Instruction Following." arXiv preprint arXiv:2011.07384 (2020).
> > [d] Blukis, Valts, et al. "Learning to Map Natural Language Instructions to Physical Quadcopter Control Using Simulated Flight." Proceedings of the Conference on Robot Learning (CoRL). 2019.
> > [f] Dosovitskiy, Alexey, et al. “An Image Is Worth 16x16 Words: Transformers for Image Recognition at Scale.” ICLR 2021: The Ninth International Conference on Learning Representations, 2021.

---

> > > ### Comment · Reviewer_kWqo · 2021-08-26
> > > **Re: Continued Response**
> > >
> > > I appreciate the detailed response from the authors.
> > >
> > > **Novelty**: I agree that, to the best of my knowledge, applying Vision Transformer is novel. For the real-world robot datasets and tasks, I am not sure if they would be available/applicable to the research community. For the simulated environment, do you plan to publicly release it?
> > >
> > > **Two-stage pipeline**: I actually do not see the change in section 5.1 addressing this.
> > >
> > > **Intermediate representations**: This paper also does not experiment with tasks that require manipulating irregular, non-rectangular, or deformable objects. While this is conceptually convincing, there are no empirical results supporting this statement. Therefore, I still believe comparing against methods that use representations such as object detection boxes would make this work more convincing.
> > >
> > > **Related work**: Five papers are published.
> > > - Modular multitask reinforcement learning with policy sketches: ICML2016
> > > - Zero-Shot Task Generalization with Multi-Task Deep Reinforcement Learning: ICML2017
> > > - Program Guided Agent: ICLR2020
> > > - Hierarchical Program-Triggered Reinforcement Learning Agents For Automated Driving: IEEE Transactions on Intelligent Transportation Systems
> > > - Reinforcement Learning of Implicit and Explicit Control Flow in Instructions: ICML 2021 (this work was arxived before the corl submission deadline but got published at ICML after the corl deadline so it makes sense that the authors do not consider it)
> > >
> > > **RL Baselines**: Sorry, I meant SPOT-Q. I am suggesting two RL baselines be included in the comparison.
> > >
> > > - Baseline 1: a model that learns to (1) predict segmentation maps with supervised learning loss using the same data used to train the proposed model, and (2) produce actions to maximize rewards. Compared to the naive RL baseline, this baseline leverages the supervision of segmentation maps, making it fairer.
> > > - Baseline 2: a multi-task RL policy that learns to solve the block rearrangement tasks with multiple configurations, similar to the training of SPOT-Q. Compared to the naive RL baseline, this baseline leverages the prior knowledge about other tasks like SPOT-Q, making it fairer.
> > >
> > > **Unseen color pairs**: holding out 6 pairs makes more sense. I wonder how holding out more portion of pairs actually increase the block accuracy from 81.6\% to 84.8\%? Can the authors comment on this?
> > >
> > > **Grid downscaling factor**: I understand that this is a hyperparameter. I am mainly concerned about not being able to learn and perform inference on varying size objects, which is a fundamental limitation of ViT itself I guess.

---

> > > > ### Author Response · Authors · 2021-08-30
> > > > **Re: Re: Continued Response**
> > > >
> > > > We appreciate your continued engagement on this paper. Please find our responses in-line below.
> > > >
> > > >
> > > > -----
> > > > > “Novelty: I agree that, to the best of my knowledge, applying Vision Transformer is novel. For the real-world robot datasets and tasks, I am not sure if they would be available/applicable to the research community. For the simulated environment, do you plan to publicly release it?”
> > > >
> > > > We plan to publicly release all datasets, tasks, and code to the community. This includes code for the simulated environment. Essentially, everything necessary to reproduce our results except for physical hardware.
> > > >
> > > > -----
> > > > > “Two-stage pipeline: I actually do not see the change in section 5.1 addressing this.”
> > > >
> > > > Our apologies for the lack of clarity here; section 5.1 lines 308-310 of the latest version addresses error propagation by noting that, when applied to multi-step tasks, small errors in our method compound over timesteps, leading to lower task completion scores.
> > > >
> > > > -----
> > > > > “Related work: Five papers are published.
> > > >
> > > > Thank you for referring us to these papers and their publication venues, we have added all of the published references to the related works section.
> > > >
> > > > -----
> > > > > “RL Baselines: Sorry, I meant SPOT-Q. I am suggesting two RL baselines be included in the comparison.
> > > >
> > > > Thanks for the additional details. We absolutely would have pursued one of these two baselines if we saw a feasible path in the context of our low-sample domain, as we did with the requested leave-additional-colors-out experiment. However, we will detail concrete related evidence below, indicating that both baselines would at this time be either infeasible, or ineffective. We would very much appreciate your consideration of this evidence, and of an alternative “perfect-masking” experiment below the “Intermediate Representations” reply, which we hope can provide a pathway towards addressing your concerns.
> > > >
> > > > > Baseline 1: a model that learns to (1) predict segmentation maps with supervised learning loss using the same data used to train the proposed model, and (2) produce actions to maximize rewards. Compared to the naive RL baseline, this baseline leverages the supervision of segmentation maps, making it fairer.
> > > >
> > > > Thank you for providing more details on this first baseline. Provided we have understood this description correctly, baseline 1 case 1 and 2 would have a segmentation at every pixel. As we understand it, such an approach is equivalent to [Zeng et al[32]’s](https://arxiv.org/pdf/1803.09956.pdf) decluttering experiments on grasping-only, and pushing and grasping. In this decluttering task they evaluate if a grasp, push, or no action (labels 0, 1, 2) will work at each pixel, which can be roughly thought of as an equivalent to a red, green, or blue block (labels 0, 1, 2). The authors’ RL method clears 100% of tables decluttered completely; the segmentation based grasp-only method degrades to 90% trial completion; and the push, grasp, failed-action method (P+G Reactive) degrades to 54.5% trial completion. The RL method by Hundt et. al.[9] that we build upon runs this same experiment, outperforming all methods in Zeng et. al.[32] by achieving 100% table clearance in fewer actions. Since a segmentation baseline approach has already been established to be ineffective for a simpler object-only case in our domain, without colors, and since a failed segmentation also cannot be updated at test time, we did not include such a baseline. Utterances, by contrast, can be updated at live test time without training thanks to the human “on the loop”, as our real robot demo attests.
> > > > We have revised the Related Work section “Q-learning for Multi-Step Tasks” to refer to these experiments.
> > > >
> > > >
> > > > > Baseline 2: a multi-task RL policy that learns to solve the block rearrangement tasks with multiple configurations, similar to the training of SPOT-Q. Compared to the naive RL baseline, this baseline leverages the prior knowledge about other tasks like SPOT-Q, making it fairer.”
> > > >
> > > > Thank you for the additional details about this second baseline. The underlying RL methods we are evaluating are not multi-task capable methods, so an experiment of this kind would itself constitute a significant amount of new work. Such baselines will also have an extremely low chance of convergence in our low-data regime with ~20,000 action and hundreds or low thousands of images, for the following reasons. First, we have obtained negative results with an end-to-end RL solution we implemented, as we mentioned previously; second, relevant experiments such as Hundt et. al.[9] Table 1 row 1, which evaluates a standard discounted Q-Learning reward, did not converge, and extending to multi-task would have a similar effect; and third, some of the latest large-scale multi-task RL research such as MT-Opt (https://arxiv.org/pdf/2104.08212.pdf) utilizes millions of time steps, years of data, and significantly more hardware to converge than we have available.
> > > >
> > > > -----
> > > >
> > > > (continued response below)

---

> > > > > ### Author Response · Authors · 2021-08-30
> > > > > **Continued response**
> > > > >
> > > > > > “Intermediate representations: This paper also does not experiment with tasks that require manipulating irregular, non-rectangular, or deformable objects. While this is conceptually convincing, there are no empirical results supporting this statement. Therefore, I still believe comparing against methods that use representations such as object detection boxes would make this work more convincing.”
> > > > >
> > > > > We agree that experiments with such objects would be interesting, and would like to address those in future work. In practice, the masks produced by the model, as visualized in Fig. 1, are contiguous rectangles made of contiguous patches, and so we feel that measuring the performance of object detection boxes may be less informative in our domain and data regime. We also refer back to our RL baseline 1 response above, which describes existing experiments from prior work that serve an equivalent purpose.
> > > > >
> > > > > We have given more thought to possible additional experiments to better examine the gap between RL performance and masking performance. Our proposal is to conduct an upper-bound perfect-masking experiment that generates an exact color-specific mask to guide the RL model to the correct object region by reading the internal simulator state. We can definitively say a perfect-masking baseline will have higher performance than our proposed method, would better ground the results found in Table 2, and we can commit to including such an experiment for the camera-ready deadline.
> > > > >
> > > > > Would conducting a perfect-masking upper-bound experiment and adding that result to Table 2 by the camera-ready deadline suffice?
> > > > >
> > > > > -----
> > > > > > “Unseen color pairs: holding out 6 pairs makes more sense. I wonder how holding out more portion of pairs actually increase the block accuracy from 81.6% to 84.8%? Can the authors comment on this?”
> > > > >
> > > > > In this case, the improvement may be due to different model selection and evaluation, since model selection for the compositionality experiments is done with a compositional held-out dev set. Given more held-out pairs, it is possible that the increased size of the dev set yielded more stable estimates of test-set performance, resulting in higher performance.
> > > > >
> > > > > -----
> > > > > > “Grid downscaling factor: I understand that this is a hyperparameter. I am mainly concerned about not being able to learn and perform inference on varying size objects, which is a fundamental limitation of ViT itself I guess.”
> > > > >
> > > > > Thanks for clarifying this concern. ViT can handle objects of varying sizes: the paper introducing ViT (ref. [a]) classifies differently-sized objects, and where ViT has been applied to semantic segmentation (refs. [b], [c]) it is also able to model objects of varying sizes
> > > > > Transformers applied to image patches operate in the same fashion as their original word sequence application in natural language processing. In either case, the sequence is broken into tokens (patches for images, words for sentences) and then has an attention mechanism that compares the tokens quadratically against each other token. Part of the strength of the transformer is in its ability to model flexible-length sequences (as an LSTM might).  Similarly, our architecture deals with varying size objects by breaking an image into patches, then the attention mechanism runs nonlinear dense transformations on those patch units, then produces a per-token classifier output. In our case, the output patches are reshaped into a mask, but there is no limit to the size of the shapes masked -- it could cover the whole image, or none of it, without any modification to the architecture.
> > > > >
> > > > >
> > > > > Thanks again for your detailed replies and your engaging interactive review.
> > > > >
> > > > > References
> > > > > -------
> > > > > [a] Dosovitskiy, Alexey, et al. "An Image is Worth 16x16 Words: Transformers for Image Recognition at Scale." International Conference on Learning Representations. 2020.
> > > > >
> > > > > [b] Strudel, Robin, et al. "Segmenter: Transformer for Semantic Segmentation." arXiv preprint arXiv:2105.05633 (2021).
> > > > >
> > > > > [c] Zheng, Sixiao, et al. "Rethinking semantic segmentation from a sequence-to-sequence perspective with transformers." Proceedings of the IEEE/CVF Conference on Computer Vision and Pattern Recognition. 2021.

---

> > > > > > ### Comment · Reviewer_kWqo · 2021-08-30
> > > > > > **Re: Continued response**
> > > > > >
> > > > > > **Novelty & two-stage pipeline & related work & grid downscaling factor**: noted and thanks.
> > > > > >
> > > > > > **RL baseline 1**: thanks for the additional explanations. While I do understand how the authors' perspective, I still feel that empirical results conducted in this exact setup are required to fully convince me. After all, this A>B and B>C so A>C statement is, while conceptually appealing, not always valid when it comes to learning systems.
> > > > > >
> > > > > > **RL baseline 2**: I am not sure how this is not possible to learn a policy conditioning on different combinations to what block to manipulate, especially in simulation. Without this baseline, it is very difficult to judge if the performance gap between the proposed method and the RL baseline should be attributed to the claim factorization (learning how to act and learning where to act) or to employing the pre-trained SPOT-Q model.
> > > > > >
> > > > > > **Intermediate representations**: I still firmly believe experimenting with different intermediate representations such as object bounding boxes is required to justify the effectiveness of using segmentation maps.
> > > > > >
> > > > > > **Perfect-masking upper-bound experiment**: this sounds like a reasonable experiment to do. However, the performance gap between this method and the proposed method would be important to justify how well the proposed method works. Without seeing the actual result, it would be difficult for me to take this into consideration.
> > > > > >
> > > > > > **Unseen color pairs**: "Given more held-out pairs, it is possible that the increased size of the dev set yielded more stable estimates of test-set performance, resulting in higher performance" sounds very strange to me. It should yield more stable estimates but should not yield higher performance. I can't help but start questioning the reliability of this experiment.
> > > > > >
> > > > > > I truly appreciate the authors' responses and the scientific rigor that they have demonstrated in the rebuttal. Yet, it seems that it is nearly impossible to resolve some of my concerns with empirical results given the little time left for the rebuttal period. I will try my best to evaluate this work based on where we stand now. I have no further questions at this point.

---

### Official Review · Reviewer_gQ8o · 2021-07-24

**Originality:** Good
**Technical Quality:** Good
**Clarity Of Presentation:** Good
**Impact:** 2

**Recommendation:**

Weak Accept: I recommend accepting the paper, but will not argue for my recommendation if the majority of other reviewers have a different opinion.

**Summary:**

This paper proposes using natural language instructions for multi-step rearrangement tasks, by mapping images and instructions to masks over pick or place locations. The model is factorized into 2 components — a pretrained SPOT-Q model independent of language, and a language-conditioned model proposed by the authors. The predictions from these two components are combined to obtain the final action.

**Issues:**

- More convincing analysis of physically feasible subset.
- Clarify whether the natural language setting is single-step or multi-step.

**Reviewer Expertise:**

Very good: Comprehensive knowledge of the area

**Strengths And Weaknesses:**

Strengths:
- Experiments are conducted both with synthetic and real images, as well as templated and crowd-sourced language. The results show that the proposed method outperforms previous methods on both visual goal prediction and on task completion metrics.
- The factorized model is interesting, and allows reusing pretrained models, thereby reducing data requirements.

Weaknesses:
- The authors discuss the advantages of using a factorized model, but do not discuss the limitations. For example, language might have more information than "where to act", like "put the red block *gently* on the blue block".
- Some analysis of how the performance changes as the no. of steps in the task is increased could be informative.
- It seems like the natural language experiments (based on the dataset from Bisk et al) involve moving only a single block. If this is true, it doesn't represent the "multi-step" rearrangement setting being addressed in the paper, and needs to be made explicit.
- The experiments on compositionality aren't particularly informative — it seems pretty intuitive that the model will learn concepts like "red block" and "blue block" independently. A more interesting question is — can the model learn to compose different object attributes? E.g. if it is only shown "small red blocks" and "large blue blocks" during training, can it generalize to "large red blocks"?
- The experiments with the physically feasible subset are not convincing — (1) are the differences statistically significant?, (2) could the difference be because the physically feasible dataset is smaller in size?

**Summary Of Recommendation:**

The proposed approach is sound, and the experimental results show that it improves over existing methods, but some additional experiments could make the paper better.

---

> ### Author Response · Authors · 2021-08-26
> **Authors’ initial gQ80 Review Response, Thank you.**
>
> Thank you for your thoughtful review and astute suggestions. We have addressed a number of them in the revised version we have uploaded, where key changes are marked with blue text.
>
> Could you please examine our updated revision pdf, with key changes in blue? We would really appreciate it!
>
> ---
>
> > 1. “The authors discuss the advantages of using a factorized model, but do not discuss the limitations. For example, language might have more information than "where to act", like "put the red block gently on the blue block".”
>
> This is an astute point, and we have added this potential limitation to the end of Section 1.
>
> ---
>
> > 2. “Some analysis of how the performance changes as the no. of steps in the task is increased could be informative.”
>
> This analysis would certainly be interesting, unfortunately, it is not feasible due to limitations in the data available for comparison, as well as the remaining time and resources available. The final task performance can only be measured on the simulated colored blocks tasks, where all tasks have the same number of steps to success. Since these experiments build on previous work and data using 4-step block stacking and row-making without NLP, their viability for significant contributions to the community is already well-established. While the Bisk data does vary in task length, its design lacks clear success conditions; so we do not measure final task performance, only per-action performance, averaged across steps. Thus, comparing performance across the number of steps in a task is infeasible.
>
> ---
>
> > 3. “It seems like the natural language experiments (based on the dataset from Bisk et al) involve moving only a single block. If this is true, it doesn't represent the "multi-step" rearrangement setting being addressed in the paper, and needs to be made explicit.”
>
> Thanks for this suggestion. We have added a clarification that the instructions are given per step, in addition to the note in section 2 that we follow Bisk et al. in formalizing multi-step tasks as Problem-Solution sequences, where multi-step tasks are broken into sequences of states paired with instructions on how to transition between states.
> The experiments on natural language data are meant to validate that the proposed masking method is viable for natural language data on multi-step tasks, as we show in Experiment 1, with a simplifying assumption that tasks should be factorized into steps. We have added a clarification to this effect to section 5.2.
>
> ---
>
> > 4. “The experiments on compositionality aren't particularly informative — it seems pretty intuitive that the model will learn concepts like "red block" and "blue block" independently. A more interesting question is — can the model learn to compose different object attributes? E.g. if it is only shown "small red blocks" and "large blue blocks" during training, can it generalize to "large red blocks"?”
>
> Thank you for bringing up this concern, our experiments investigate if the model is learning “red” and “block” independently, as compared to learning “red block” and “blue block” independently. To prevent confounding compositionality with sparsity, we analyze the compositionality of the learned model using single adjective-object pairings, as is standard in linguistic practice, cf. [e].
> To ensure we are understanding your suggestion correctly, could you please provide more details on why adding additional adjectives would make the analysis more interesting? Could you please also elaborate on why you think it is intuitive that the model would learn to compose color terms with blocks, rather than simply memorizing all block combinations?
> ---
>
> > 5. “The experiments with the physically feasible subset are not convincing — (1) are the differences statistically significant?, (2) could the difference be because the physically feasible dataset is smaller in size?”
>
> Thanks for pointing this out. We have added statistical testing (in blue) to this table which indicates that the results are statistically significant, and not due to the smaller size of the dataset. More details on the filtered subset of the dataset have been added to Appendix C2.
>
> [e] Partee, Barbara. "Compositionality and coercion in semantics: The dynamics of adjective meaning." Cognitive foundations of interpretation 2007 (2007): 145-161.

---

> > ### Comment · Reviewer_gQ8o · 2021-08-28
> > **Clarification on compositionality**
> >
> > Thanks for addressing most of the comments.
> >
> > Regarding compositionality: From what I understand, I don't think the current experiments investigate whether the model is learning "red" and "block" independently — isn't the model always getting instructions of the form "move/stack the __ block next to/on the __ block"? If so, the word "block" is essentially redundant, since it's invariant. It seems for the compositionality experiments, the training data doesn't contain one pair of objects, like red block and blue block, and is then tested on that pair. So, the experiments show whether the model is learning "red block" and "blue block" as independent concepts or not.
> > One way to think about additional adjectives is what are we trying to test compositionality on? If it is just linguistic phrases, then the current experiments are okay. But another possibility is compositionality in language grounding. Since this paper involves language grounding (as opposed to ref [e]), the more interesting compositionality question in my opinion is — if it is shown "small red blocks", does it learn to *ground* the meaning of the words "small" and "red" independently?

---

> > > ### Author Response · Authors · 2021-08-30
> > > **Clarification on compositionality**
> > >
> > > Thank you for this astute observation -- you are correct in pointing out that since “block” occurs without syntactic variation, the compositionality is not between the adjective and noun, but rather at the level of the instruction, with the meaning of “red” remaining invariant across contexts like “stack the red block on the blue block” and “stack the green block on the red block”. We have uploaded a revision (lines 185-196) clarifying that the model is learning to interpret colors separately (i.e. interpreting the meaning of the utterance compositionally) rather than interpreting each instruction type as an unrelated atom. If we consider the input as an abstracted ternary function `place(color_a, on, color_b)` we are able to generalize from, for example, `place(red, on, blue)` and `place(blue, on, green)` to `place(red, on, green)` even though `place(red, on, green)`, and `place(green, on, red)` have not been seen at train time.
> > > We agree that showing that the model can ground “small” and “red” independently and compositionally would be interesting. While our current experiments show that the model learns to ground color terms separately, we currently can only speculate on whether it would be able to learn to separate additional modifiers into concepts. Unfortunately our dataset does not contain different block sizes, and so in light of the resources and time available to us we must leave this item to future work.
> > >
> > > Your framing of the distinction between linguistic and grounded compositionality brings up an interesting point: our current framework in fact unifies grounded and linguistic compositionality in our domain. The linguistic interpretation of compositionality (as in [e]) typically centers around a symbolic meaning representation of an utterance (e.g. lambda calculus). In our system, the pair of masks produced by the model from an input utterance can themselves be seen as (albeit domain-limited) meaning representations, while also being an explicit mapping grounding the color terms in the utterance to a visual input. In other words, when our model produces a grasp mask around a red block from the input “place the red block on the blue block”, it is simultaneously producing a representation of what the input means and grounding the color adjectives directly to the input. Based on the fact that the model is able to generalize to unseen color pairs, the output reflects compositional reasoning over the input utterance, both when viewed as a meaning representation and a grounded mapping.
> > >
> > > Thanks for your continued interest and engagement on this point, it has made for an exciting and enriching review process.

---

### Official Review · Reviewer_h8PP · 2021-07-24

**Originality:** Good
**Technical Quality:** Very Good
**Clarity Of Presentation:** Very Good
**Impact:** 3

**Recommendation:**

Weak Accept: I recommend accepting the paper, but will not argue for my recommendation if the majority of other reviewers have a different opinion.

**Summary:**

This work leverages recent vision models to perform natural language guided block stacking.  They use synthetic language and a real crowdsourced resource (filtered for physical viability).  Limited experiments (due to covid) are performed on a physical manipulator.  The task requires understanding spatial language expressions for both identifying the referent and target location.

**Issues:**

The primary limitation of the current work is limited error analysis which makes it difficult to diagnose what the next steps are for the research community and ideally more physical experiments (though this is not a limitation that can be expected to be addressed in the current climate.)

**Reviewer Expertise:**

Very good: Comprehensive knowledge of the area

**Strengths And Weaknesses:**

**Strengths**
1. Novel model architecture for the task with good comparison baselines
2. Analysis of simple stacking and rows in both synthetic and real images.
3. Curation of the Bisk et al. dataset for robotic tasks and evaluation on this real language setting
4. Manipulation performed with simulated UR-5
5. Basic manipulation demonstration of simplest setting on physical manipulator

**Weaknesses**
1. Insufficient analysis of the difficulties of real language (e.g. behavior in abstract language settings)
2. Limited physical manipulator experiments which also means limited analysis of the causes for the model's transfer errors to the real setting.

**Summary Of Recommendation:**

The authors present and demonstrate the utility of novel models for language guided control/actions based on recent advances.  They do basic comparisons against language and task complexity and include experiments both on simulated and real control platforms.

---

> ### Author Response · Authors · 2021-08-26
> **Authors’ initial h8PP Review Response, Thank you.**
>
> Thank you for your insightful comments. Your suggestions, such as including additional error analysis, have served to improve our submission.
>
> Could you please examine our updated revision pdf with key changes in blue? We would really appreciate it!
>
> ---
> > 1. “Insufficient analysis of the difficulties of real language (e.g. behavior in abstract language settings)”
>
> Thank you for pointing this out. We have added a qualitative analysis of errors to the supplementary materials. These errors deal primarily with abstract linguistic settings, as well as complex phenomena such as ellipsis and coreference. The analysis can be found in our revised submission, where we have highlighted it in blue.
>
> ---
> > 2. “Limited physical manipulator experiments which also means limited analysis of the causes for the model's transfer errors to the real setting [...] though this is not a limitation that can be expected to be addressed in the current climate”
>
> We agree with this point, and would have done more extensive real robot experiments without the barriers imposed by COVID-19.  Nonetheless, we have worked to ground our experiments as realistically as is feasible via the real data we have available.
>
> Thanks again for your consideration and for reviewing our submission.

---

### Official Review · Reviewer_1smN · 2021-07-26

**Originality:** Good
**Technical Quality:** Very Good
**Clarity Of Presentation:** Good
**Impact:** 3

**Recommendation:**

Weak Accept: I recommend accepting the paper, but will not argue for my recommendation if the majority of other reviewers have a different opinion.

**Summary:**

The main contributions of this paper could be concluded as following:
(1) Proposed a transformer-based model to build mapping from natural language instructions to real scenes with pixel-wise Q values and masks which are compatible with both supervised learning and reinforcement learning methods.
(2) Apply the proposed transformer-based model on multi-step rearrangement tasks and compare it with a baseline model:Unet-based VGP model in metrics of block acc and IES
(3) Give out a dataset based on Bisk dataset mentioned in paper with both real and simulated samples to for training model for multi-step rearrangement task in more realistic scenes


**Issues:**

1.	Explainability of propose pixel-wise mapping method
2.	A more comprehensive validation of effiency of propose transformed method


**Reviewer Expertise:**

Good: General knowledge of the area

**Strengths And Weaknesses:**

Strengths:
1.	Innovatively proposed the method to mapping input image with natural language instructions with pixel-wise Q-value and masks which is feasible in practice and useful
2.	The proposed model was validated that it can converge fast with much fewer samples than traditional transformer model in training phase.
3.	Design a complete experiment setting to compare propose model with baseline in aspects of training efficiency, genralization toward different taks types et al.
Weakness:
1.	The mapping lack explainability, a theoritical explaination or citiation should be presented to clear the reasonability of propose mapping method
2.	Data ablation experiments should be more comprehensive. Fisrt, generalizability should be taken into consideration, a different dataset could be used to validate the propose model can still improve fast or not. Second, robustness should be validated. For example, add some noisy into train set.
3.	The released dataset need to validated in several aspects such as data type balance,reasonablity of insturctions and size


**Summary Of Recommendation:**

I recommend this paper for its contributions on proposing  a transformer-based model to solve multi-step rearrangement task in realisitc simulated environment as it provides an innoative and feasible method to map natrual language instructions to input scene images and the experiments shows that it could be more effective and efficient in training than most transformer-based models.

---

> ### Author Response · Authors · 2021-08-21
> **Clarification RE explainability and robustness**
>
> Firstly, thank you for your detailed and thoughtful review. To best respond to the concerns you pointed out, we wanted to ensure that we understand them correctly.
> Specifically, in point 4, you mention the mapping lacks explainability. If it is not too much of an imposition, could we ask you to perhaps elaborate on this? By mapping, I assume you're referring to the mapping between images and Transformer-produced segmentation masks, rather than the mapping between inputs and Q-values. Is that correct? With respect to a theoretical explanation or citation, would that be an explanation clarifying how the mapping between language and segmentation masks can be used to explain model behavior, or a citation to that effect?
>
> With regards to point 5 (generalizability), I would like to start by pointing out that we do in fact validate our method on two datasets, the first being the colored blocks domain, and the second one being the Bisk dataset. In the first domain, we apply our method to both simulated and real images. Could you provide additional details on what kind of dataset, beyond the ones explored here, would assuage your concerns about the method's generalizability?
> In addition, as mentioned in Section 4.1, in the simulated colored blocks domain we do in fact add Gaussian noise during train time as a data augmentation step. Does this serve to address your suggestion to add noise to the train set? What result here would convince you of the method's robustness to noise?

---

> ### Author Response · Authors · 2021-08-26
> **Authors’ initial 1smN Review Response, Thank you.**
>
> Thank you for your insightful comments. We first refer you to our metareview response, which addresses several very important concerns you raise, specifically 2 and 3, and we respond to the remaining concerns below.
>
> Could you please examine our updated revision pdf with key changes in blue? We would really appreciate it!
>
> ---
> > “1. The mapping lack explainability, a theoritical explaination or citiation should be presented to clear the reasonability of propose mapping method”
>
> We appreciate your concern, but we do not make any major claims about the explainability of our method, noting only that a user can interpret, i.e. read a chart like fig.1, and visually infer if an action like a grasp is reasonable or not at that location. This is consistent with Hundt et al[9], Zeng et al[23], among others.
>
> Could you elaborate on the concern regarding theoretical explanation so we might better address it?
>
> ---
> > “2. Data ablation experiments should be more comprehensive. Fisrt, generalizability should be taken into consideration, a different dataset could be used to validate the propose model can still improve fast or not. Second, robustness should be validated. For example, add some noisy into train set.”
>
> Thank you for noting your concerns, we have addressed this item in our metareview reply, so we would appreciate it if you could look there.
>
> ---
> > “3. The released dataset need to validated in several aspects such as data type balance,reasonablity of insturctions and size”
>
> Thank you for this suggestion. We have added a table with details in Appendix C2.
>
> Thanks again for your feedback and taking the time to review our submission.

---

> > ### Comment · Reviewer_1smN · 2021-09-03
> > **Thanks for the response.**
> >
> > Thanks for the response. The response answers some of my concerns, and I will keep my original rating.

---

### Public Comment · (anonymous) · 2021-08-31
**Lack of novelty in using pre-trained language models to lower data requirements.**

I would like to point out that the use of pre-trained language models to overcome annotation requirements when following crowd-sourced natural language instructions is not a novel contribution of this work.

Two recent examples which employ this technique with the same motivation are:
https://arxiv.org/abs/2005.09382, https://arxiv.org/abs/2005.07648.

In the response, you cite "ability to reuse pre-trained models (reviewer gQ8o)" as a source of novelty. It might be helpful to clarify that only the application of pre-trained *control* models (not language models) is novel here.

---

> ### Author Response · Authors · 2021-09-01
> **Re: using pre-trained language models**
>
> Thank you for your interest in the paper. It is indeed the case that our model uses pre-trained models both for control and in some instances for language feature extraction. We are of course not the first work to use pre-trained masked language models (e.g. BERT) for linguistic feature extraction, and did not intend to have our statement interpreted in that way. Additional clarification might be found in the context of the original review by reviewer gQ8o as well as the paper itself, where the distinction is made explicit (cf. lines 45-47, 140-143). When the paper and whole review are taken together in context, it is our hope that our claims about the novelty of our work are clear, and are not taken to include the claim that we are using BERT in a novel way.
>
> Furthermore, when we use pre-trained control models (Section 5.1) we in fact do not use any pre-trained language representations, instead learning embeddings from scratch, and when we do use BERT (Section 5.2) we evaluate the masking model alone, with no control model.

---

### Meta-Review · Area_Chair_sf66 · 2021-08-03

**Recommendation:** Accept (Poster)
**Confidence:** 4

**Metareview:**

After the author response and the discussion phase, the reviewers agree that there is room for improvement, but appreciate both the key idea of the paper of using pre-trained models to efficiently learn language-conditioned tasks and the effort of demonstrating results going from real language all the way to low-level robot control. The results on a real robot are also appreciated and make it particularly well-suited for CoRL. I recommend accept.

--------------------------------------

This paper seems to be somewhat borderline. It presents an interesting method to a relevant problem, includes some ablations to understand the importance of different design choices, and includes experiments with natural language along with some experiments on a real manipulator.

The paper also has several weaknesses, including the following:
* The experiments are limited to block re-arrangement tasks, rather than a more general class of rearrangement tasks. It does not seem like the method is limited to such simple block-based problems, but the experiments do not validate its generality.
* There are a number of experiments that would strengthen the empirical analysis, including a stronger RL baseline, more analysis on the failure cases with natural language, and more experiments on a real robot. (see comments from reviewers kWqo and h8PP). Including this more detailed analysis and comparisons would make it possible for the reader to identify/understand important future steps in this research area.
* The related work section does not discuss the relation of this work to a number of relevant works. Beyond the works mentioned by reviewer kWqo, there are additional prior works that study rearrangement & manipulation tasks from language instructions [1,2,3].

The final result of the paper will depend on the author response and reviewer discussion. If some of the weaknesses can be addressed, then it will most likely be above the bar.

* [1] Learning to Understand Goal Specifications by Modeling Reward. ICLR 2019.
* [2] Language as an Abstraction for Hierarchical Deep Reinforcement Learning. NeurIPS 2019.
* [3] Language Conditioned Imitation Learning over Unstructured Data. arXiv 2020 / RSS 2021

---

> ### Author Response · Authors · 2021-08-26
> **Authors’ initial Metareview response**
>
> Thank you for your thoughtful assessment, we have revised our submission in accordance with this feedback.
>
> Could the reviewers please examine our updated revision pdf with key changes in blue? We would really appreciate it!
>
> We will divide our responses by topic. Numbered cites will refer to the pdf and lettered cites will be inline below.
>
> ---
> > 1. “The experiments are limited to block re-arrangement tasks, rather than a more general class of rearrangement tasks. It does not seem like the method is limited to such simple block-based problems, but the experiments do not validate its generality.”
>
> We appreciate that the specific metrics of generality is considered a matter of paramount concern for most submissions. To this end, our method operates with utterances and raw pixels as input that then maps to very fine-grained discrete actions as output. We operate without access to keypoints, bounding boxes, object models, canonical poses, or other similar assumptions about the physical structure of the objects in the scene. The specific RL models we build upon are also already proven to be capable of grasping a variety of shapes in both simulated and real environments [9,16]. Many Transformer-based methods, such as [a], already perform very well on standard segmentation benchmarks, while other work addresses transformers for shape-related problems [b], so we can already expect our masking approach is highly likely to prove effective with no changes beyond very minor adjustments.
>
> Our methods already make significant progress on other challenging metrics of generalization that include real robot data, simulated robot data, human utterances, multi-step tasks, and compositionality. We already experiment with real utterances in physically realistic environments not available in prior work, and will make the data available to the community. For these reasons, we would appreciate it if we can note in our paper that experiments with respect to a shape-based generality metric could serve as part of a natural future extension.
>
> ---
> > 2. “experiments that would strengthen the empirical analysis, including a stronger RL baseline” and reviewer kWqo suggests “(1) learning the RL policy with an auxiliary task of predicting segmentation maps”.
>
> We actually ran exactly this sort of experiment prior to submission, but the model did not converge. To address this concern we now note in section 5.1 that “We also ran a small end-to-end Q-Learning experiment that backpropagates through both the FCN and Transformer, but the model did not converge.” Could you please let us know if this addresses your concern?
> The RL model that serves as part of our method [9] is, to the best of our knowledge, state of the art for published RL models in the domains we examine, so no other stronger RL baseline is available. Known alternative RL methods for multi-step tasks are intractable on a single GPU workstation [9], which is the type of hardware we have.
> ---
> > 3. “strengthen the empirical analysis” with “more analysis on the failure cases with natural language”
>
> Thank you for this suggestion, we have incorporated an additional qualitative error analysis on the natural language data into our paper in Appendix C.4, which we have highlighted in blue.
>
> ---
>
> > 4. strengthen the empirical analysis" with more experiments on a real robot.
>
> Unfortunately, our real experiments have been severely curtailed due to COVID-19 restrictions, as we note in the appendix, so we would appreciate cognizance of this factor that remains outside of our control.
>
> ---
>
> > 5. (see comments from reviewers kWqo and h8PP).
>
> Our replies to additional specific comments will go below their reviews.
>
> ---
> > 6. “The related work section does not discuss the relation of this work to a number of relevant works. Beyond the works mentioned by reviewer kWqo, there are additional prior works that study rearrangement & manipulation tasks from language instructions [1,2,3].”
>
> Thank you for these suggestions, we have incorporated them into Section 2, Related Work.
>
> Thanks again for taking the time and effort you have put into reviewing our submission.
>
> References
> ----
>
> [a] Zheng, Sixiao, et al. “Rethinking Semantic Segmentation from a Sequence-to-Sequence Perspective with Transformers.” Proceedings of the IEEE/CVF Conference on Computer Vision and Pattern Recognition, 2020, pp. 6881–6890.
>
> [b] M. C. H. Lee, K. Petersen, N. Pawlowski, B. Glocker and M. Schaap, "TeTrIS: Template Transformer Networks for Image Segmentation With Shape Priors," in IEEE Transactions on Medical Imaging, vol. 38, no. 11, pp. 2596-2606, Nov. 2019, doi: 10.1109/TMI.2019.2905990.

---

> > ### Comment · Area_Chair_sf66 · 2021-08-27
> > **Response**
> >
> > Thank you for the response.
> >
> > **Generality**. To be clear, I completely agree that the method is likely applicable to objects beyond blocks. My comment is that the paper would be stronger if it included experiments that have a greater variety of objects.
> >
> > **RL Baseline.** See reviewer kWqo's reply to your response.
> >
> > **Empirical analysis.** Acknowledged.
> >
> > **Related work.** The changes to the related work include multiple incorrect statements:
> > * Jiang et al. [21] is not performing navigation, nor is it performing imitation learning.
> > * Lynch & Sermanet [27] include experiments from pixel observations and perform imitation learning, not RL.
> > * Papers [14] and [32] do not perform "formal RL" (Formal RL is not a term that exists to my knowledge.)
> >
> > It seems possible that you may have accidentally swapped the citations [21] and [27]. However, that would still be inaccurate since Jiang et al. [21] also includes experiments from pixel observations and [27] also does not perform navigation. Please make any future changes to the paper with more care. My role as AC is not to line edit your paper.
> >
> > Slightly less importantly, the related work section primarily comments on how the experiments of this paper differs from the experiments in prior works. It would be much more informative to comment on how the proposed paper differs from these prior works at a conceptual level. Finally, also see reviewer kWqo's response for published papers that you missed.

---

> > > ### Author Response · Authors · 2021-08-27
> > > **Updated revision**
> > >
> > > We’re getting used to the quick turnaround process and we updated too soon. We apologize for the incorrect statements in the previous revision. We have uploaded a revision with an updated related works section, which addresses the papers suggested in the metareview, as well as all of the published works from reviewer kWqo's comments.
> > >
> > > We will be following up on reviewers' replies in their respective threads. Thank you for your time and your valuable input.

---

### Decision · Program_Chairs · 2021-09-13

**Decision:**

Accept (Poster)

**Comment:**

After the author response and the discussion phase, the reviewers agree that there is room for improvement, but appreciate both the key idea of the paper of using pre-trained models to efficiently learn language-conditioned tasks and the effort of demonstrating results going from real language all the way to low-level robot control. The results on a real robot are also appreciated and make it particularly well-suited for CoRL. I recommend accept.

--------------------------------------

This paper seems to be somewhat borderline. It presents an interesting method to a relevant problem, includes some ablations to understand the importance of different design choices, and includes experiments with natural language along with some experiments on a real manipulator.

The paper also has several weaknesses, including the following:
* The experiments are limited to block re-arrangement tasks, rather than a more general class of rearrangement tasks. It does not seem like the method is limited to such simple block-based problems, but the experiments do not validate its generality.
* There are a number of experiments that would strengthen the empirical analysis, including a stronger RL baseline, more analysis on the failure cases with natural language, and more experiments on a real robot. (see comments from reviewers kWqo and h8PP). Including this more detailed analysis and comparisons would make it possible for the reader to identify/understand important future steps in this research area.
* The related work section does not discuss the relation of this work to a number of relevant works. Beyond the works mentioned by reviewer kWqo, there are additional prior works that study rearrangement & manipulation tasks from language instructions [1,2,3].

The final result of the paper will depend on the author response and reviewer discussion. If some of the weaknesses can be addressed, then it will most likely be above the bar.

* [1] Learning to Understand Goal Specifications by Modeling Reward. ICLR 2019.
* [2] Language as an Abstraction for Hierarchical Deep Reinforcement Learning. NeurIPS 2019.
* [3] Language Conditioned Imitation Learning over Unstructured Data. arXiv 2020 / RSS 2021